# Damped Anderson Mixing for Deep Reinforcement Learning: Acceleration, Convergence, and Stabilization

**Ke Sun**[*1], **Yafei Wang**[*1], **Yi Liu**[1], **Yingnan Zhao**[1,2], **Bo Pan**[1], **Shangling Jui**[3],
**Bei Jiang**[1], **Linglong Kong**[1†]
[1]University of Alberta, Edmonton, Canada
[2]Harbin Institute of Technology, Harbin, China
[3]Huawei Technologies Ltd.
{ksun6,yafei2,yliu16,yingnan6,pan1,bei1,lkong}@ublberta.ca
jui.shangling@huawei.com

## Abstract

Anderson mixing has been heuristically applied to reinforcement learning (RL) algorithms for accelerating convergence and improving the sampling efficiency of deep RL. Despite its heuristic improvement of convergence, a rigorous mathematical justification for the benefits of Anderson mixing in RL has not yet been put forward. In this paper, we provide deeper insights into a class of acceleration schemes built on Anderson mixing that improve the convergence of deep RL algorithms. Our main results establish a connection between Anderson mixing and quasi-Newton methods and prove that Anderson mixing increases the convergence radius of policy iteration schemes by an extra contraction factor. The key focus of the analysis roots in the fixed-point iteration nature of RL. We further propose a stabilization strategy by introducing a stable regularization term in Anderson mixing and a differentiable, non-expansive MellowMax operator that can allow both faster convergence and more stable behavior. Extensive experiments demonstrate that our proposed method enhances the convergence, stability, and performance of RL algorithms.

## 1 Introduction

In reinforcement learning (RL) [28], an agent seeks an optimal policy in a sequential decision-making process. Deep RL has recently achieved significant improvements in a variety of challenging tasks, including game playing [20, 26, 18] and robust navigation [19]. A flurry of state-of-the-art algorithms have been proposed, including Deep Q-Learning (DQN) [20] and variants such as Double-DQN [13], Dueling-DQN [31], Deep Deterministic Policy Gradient (DDPG) [16], Soft Actor-Critic [12] and distributional RL algorithms [5, 17, 33], all of which have successfully solved end-to-end decision-making problems such as playing Atari games. However, the slow convergence and sample inefficiency of RL algorithms still hinders the progress of RL research, particularly in high-dimensional state spaces where deep neural network are used as function approximators, making learning in real physical worlds impractical.

To address these issues, various acceleration strategies have been proposed, including the classical Gauss-Seidel Value Iteration [23] and Jacobi Value Iteration [24]. Another popular branch of techniques accelerates RL by leveraging historical data. Interpolation methods such as Average-DQN [2]

---

[*]Equal contributions in alphabetical order
[†]Corresponding author

35th Conference on Neural Information Processing Systems (NeurIPS 2021).

have been widely used in first-order optimization problems [6] and have been proven to converge faster than vanilla gradient methods. As an effective multi-step interpolation method, Anderson mixing [30, 9], also known as Anderson acceleration, has attracted great attention from RL researchers. The insight underpinning of Anderson acceleration is that RL [28] is intrinsically linked to fixed-point iterations: the optimal value function is the fixed point of the Bellman optimality operator. These fixed-points are computed recursively by repeatedly applying an operator of interest [11]. Anderson mixing is a general method to accelerate fixed-point iterations [30] and has been successfully applied to fields, such as the computational chemistry [22] or electronic structure computation [1]. In particular, Anderson acceleration leverages the $m$ previous estimates in order to find a better estimate in a fixed-point iteration. To compute the mixing coefficients in Anderson iteration, it searches for a point with a minimal residual within the subspace spanned by these estimates. It is thus natural to explore the efficacy of Anderson acceleration in RL settings.

Several works [15, 25] have attempted to apply Anderson acceleration to reinforcement learning. Anderson mixing was first applied to value iteration in [11, 15] and resulted in significant convergence improvements. Regularized Anderson acceleration [25] was recently proposed to further accelerate convergence and enhance the final performance of state-of-the-art RL algorithms in various experiments. However, previous applications of Anderson acceleration were typically heuristic: consequently, these empirical improvements in convergence have so far lacked a rigorous mathematical justification.

In this paper, we provide deeper insights into Anderson acceleration in reinforcement learning by establishing its connection with quasi-Newton methods for policy iteration and improved convergence guarantees under the assumptions that the Bellman operator is differential and non-expansive. MellowMax operator is adopted to replace the max operator in policy iteration to simultaneously guarantee faster convergence of value function and reduce the estimated gradient variance to yield stabilization. In addition, we analyze the stability properties of Anderson acceleration in policy iteration and propose a stable regularization to further enhance the stability. These key two factors, i.e., the stable regularization and the theoretically-inspired MellowMax operator, are the basis for our *Stable Anderson Acceleration (Stable AA)* method. Finally, our experimental results on various Atari games demonstrate that our Stable AA method enjoys faster convergence and achieves better performance relative to existing Anderson acceleration baselines. Our work provides a unified analytic framework that illuminates Anderson acceleration for reinforcement learning algorithms from the perspectives of acceleration, convergence, and stabilization.

## 2 Acceleration and Convergence Analysis of Anderson Acceleration on RL

We first present the notion of Anderson acceleration in the reinforcement learning and then provide deeper insights into the acceleration if affords by establishing a connection with quasi-Newton methods. Finally, a theoretical convergence analysis is provided to demonstrate that Anderson acceleration can increase the convergence radius of policy iteration by an extra contraction factor.

**Background**   Consider a Markov decision process (MDP) specified by the tuple $(\mathcal{S}, \mathcal{A}, R, p, \gamma)$, where $\mathcal{S}$ is a set of the states and $\mathcal{A}$ is a set of actions. The functions $R : \mathcal{S} \times \mathcal{A} \to \mathbb{R}$ and $p : \mathcal{S} \times \mathcal{A} \times \mathcal{S} \to [0, 1]$ are the reward function, with $R_t = R(s, a)$, and transition dynamics function, respectively for the MDP. The discount rate is denoted by $\gamma \in [0, 1)$ and determines the relative importance of immediate rewards relative to rewards received in the future. The $Q$-value function evaluates the expected return starting from a given state-action pair $(s, a)$, that is, $Q^\pi(s, a) = \mathbb{E}\left[\sum_{t=0}^{\infty} \gamma^t R_{t+1} \mid s_0 = s, a_0 = a\right]$. A policy $\pi(a|s)$ is a distribution mapping the state space $\mathcal{S}$ to the action space $\mathcal{A}$.

### 2.1 Anderson Acceleration in Policy Iteration

We focus on the tabular case to enable the theoretical analysis of Anderson acceleration in value (policy) iteration, which can be naturally applied to function approximation. Both the value iteration ($V$-notation) and the policy iteration ($Q$-notation) can have Anderson acceleration applied to them to improve convergence. However, theoretical analysis has shown that value iteration enjoys a $\gamma$-linear convergence rate, i.e., $\|V^{(t)} - V^*\|_\infty \leq \gamma\|V^{(t-1)} - V^*\|_\infty$, where $V^{(t)}$ is the value function in the iteration step $t$ and $V^*$ is the optimal value function, while policy iteration converges

faster. This is due to the fact that policy iteration more fully evaluates the current policy than does value iteration. Additionally, policy iteration is more fundamental and scales more readily to deep reinforcement learning. For this reason, we analyze our method in the policy iteration setting under $Q$-notation as value iteration is a special case of policy iteration. Thus, our analysis also applies under value iteration.

We first focus on the control setting where the optimal value of state-action pair $Q^*(s, a)$ is defined recursively as a function of the optimal value of the other state-action pair:

$$Q^*(s, a) = R(s, a) + \gamma \sum_{s' \in S} p\left(s' \mid s, a\right) \cdot \max_{a'} Q^*\left(s', a'\right). \tag{1}$$

Combining policy evaluation and policy improvement, the resulting policy iteration algorithm is equivalent to solving for the fixed point of the Bellman optimality operator $\mathcal{T} : \mathbb{R}^{|S \times \mathcal{A}|} \to \mathbb{R}^{|S \times \mathcal{A}|}$ with $\mathcal{T}Q(s, a) = R(s, a) + \gamma \sum_{s' \in S} p\left(s'|s, a\right) \cdot \max_{a'} Q\left(s', a'\right)$.

As a general technique to speed up fixed-point iteration [30], Anderson acceleration has been successfully yet heuristically applied to reinforcement learning algorithms [25, 11, 15]. Specifically, Anderson acceleration linearly combines the $m$ previous estimates to yield a better iteration target in the fixed point iteration. Geometrically, Anderson acceleration applies the operator to a point that has a minimal residual within the subspace spanned by these estimates. In policy iteration, Anderson acceleration maintains a memory of the previous $Q$ function values and updates the iterate as a linear combination of these values with dynamic weights $\alpha^{(k)}$ in the $k$th iteration step. Specifically,

$$Q^{(k+1)}(s, a) = (1 - \beta_k) \sum_{i=0}^{m} \alpha_i^{(k)} Q^{(k-m+i)}(s, a) + \beta_k \sum_{i=0}^{m} \alpha_i^{(k)} \mathcal{T}Q^{(k-m+i)}(s, a), \tag{2}$$

where $0 \leq \beta_k \leq 1$ is the damping parameter. All of the coefficients $\alpha_i^{(k)}$ in the coefficient vector $\alpha^{(k)}$ are computed following

$$\alpha^{(k)} = \underset{\alpha \in \mathbb{R}^{m+1}}{\operatorname{argmin}} \left\| \sum_{i=0}^{m} \alpha_i \left( \mathcal{T}Q^{(k-m+i)} - Q^{(k-m+i)} \right) \right\|_2 = \underset{\alpha \in \mathbb{R}^{m+1}}{\operatorname{argmin}} \left\| {\Delta_k'}^T \cdot \alpha \right\|_2, \text{ s.t. } \sum_{i=0}^{m} \alpha_i = 1, \tag{3}$$

where ${\Delta_k'}^T = [e_{k-m}, \cdots, e_k] \in \mathbb{R}^{|S \times \mathcal{A}| \times (m+1)}$ and $e_k = \mathcal{T}Q^{(k)} - Q^{(k)} \in R^{|S \times \mathcal{A}|}$ is the Bellman residuals matrix. By the Karush-Kuhn-Tucker conditions, the analytic solution of optimal coefficient vector $\alpha^k$ is

$$\alpha^{(k)} = \frac{\left( \Delta_k' {\Delta_k'}^T \right)^{-1} \mathbf{1}}{\mathbf{1}^T \left( \Delta_k' {\Delta_k'}^T \right)^{-1} \mathbf{1}}, \tag{4}$$

where $\mathbf{1}$ denotes the vector with all components equal to one.

## 2.2 Connection Between Damped Anderson Acceleration and Quasi-Newton Methods

We know that the optimization problem is closely linked with solving a fixed-point iteration problem by directly solving its first-order condition. Inspired by [8], we show that Anderson acceleration in policy iteration attempts to perform a special form of quasi-Newton iteration from its optimization problem behind.

To illuminate this connection, we firstly show that the original constrained optimization to obtain the optimal $\alpha^{(k)}$ in Eq.(3) can be equivalent to the unconstrained one

$$\tau^{(k)} = \operatorname{argmin}_{\tau \in \mathbb{R}^m} \|e_k - H_k \tau\|^2, \tag{5}$$

where we let $n = |S| \times |A|$, and then $H_k = [e_k - e_{k-1}, \cdots, e_{k-m+1} - e_{k-m}] \in \mathbb{R}^{n \times m}$. $\tau^{(k)} = \left[ \tau_0^{(k)}, \tau_1^{(k)}, \cdots, \tau_{m-1}^{(k)} \right]^T \in \mathbb{R}^m$ with $\tau_i^{(k)} = \sum_{j=0}^{m-i-1} \alpha_j^{(k)}$. Let $\delta_k = Q^{(k)} - Q^{(k-1)}$, $\Delta_k = [\delta_k, \delta_{k-1}, \cdots, \delta_{k-m+1}] \in \mathbb{R}^{n \times m}$. We show that the updating rule of $Q(s, a)$ in Anderson acceleration can be expressed as a quasi-Newton form in Proposition 1.

**Proposition 1.** *By conducting the damped Anderson acceleration (Eq.(2) and (3)) on the policy iteration, the updating of $Q^{(k+1)}$ can be reformulated as*

$$Q^{(k+1)} := Q^{(k)} - G_k e_k \tag{6}$$

*where $G_k = (\Delta_k + \beta_k H_k)\left(H_k^T H_k\right)^{-1} H_k^T - \beta_k I$ can serve as an approximation of inverse Jacobian matrix of $e_k = TQ^{(k)} - Q^{(k)}$, and $I$ is an identity matrix.*

Proposition 1 points out that Anderson acceleration on policy iteration additionally leverages more information about the fixed-point residual $e_k = TQ^{(k)} - Q^{(k)}$ to update the $Q$ function. Particularly, the first part $(\Delta_k + \beta_k H_k)(H_k^T H_k)^{-1} H_k^T$ in $G_k$ contains partial structure matrix information about the real inverse of Jacobian matrix, which has the huge potential to speed up the convergence of the fixed-point iteration. More importantly, the results established in [30] and [9] can been seen as special cases of Proposition 1 with $\beta_k = 1$. If we directly get rid of the first part in $G_k$ and set $\beta_k = 1$, the updating rule exactly degenerates to the Q-value function iteration without Anderson acceleration.

### 2.3  Convergence Rate Analysis of Anderson Acceleration on RL

The success of the Anderson acceleration to reduce the residual is coupled in the algorithm iteration at each stage. Let $e_k^\alpha = \sum_{j=0}^m \alpha_j^{(k)} (TQ^{(j)} - Q^{(j)})$. The stage-$k$ gain $\theta_k$ can be defined by $\|e_k^\alpha\|_\infty = \theta_k \|e_k\|_\infty$. As $\alpha_k^{(k)} = 1, \alpha_j^{(k)} = 0, j \neq k$, i.e., $m = 0$, is an admissible solution to the optimization problem in Eq. (3), it immediately follows that $0 \leq \theta_k \leq 1$. The key to rigorously show that Anderson acceleration can speed up the convergence of policy iteration by taking a linear combination of history steps is connecting the gain $\theta_k$ to the differences of consecutive iterates $Q^{(k)}$ and residual terms $e_k$. As discussed in the following part, the improvement of the convergence rate of the policy iteration by using the acceleration technique is characterized by $\theta_k$. We first consider the following assumption about the operator $\mathcal{T}$ used to guarantee the first and second order derivatives of $\mathcal{T}$ are bounded, as in [7].

**Assumption 1.** *Assume the Bellman operator $\mathcal{T}$ acting on state-action value function $Q$ has a fixed point $Q^*$, and there are positive constants $c_1$ and $c_2$ such that*

*1. $\mathcal{T} \in C^2(\mathbb{R}^{|\mathcal{S} \times \mathcal{A}|})$.*

*2. The first derivative of $\mathcal{T}$ is bounded by $c_1$.*

*3. The second derivative of $\mathcal{T}$ is bounded by $c_2$.*

**Theorem 1.** *Under Assumption 1, if the coefficients $\alpha_i^{(k)}$ remain bounded and away from zero, the following bound holds for the fixed point residual $e_k$ from Eq. (2) and (3) with depth $m$*

$$\|e_k\|_\infty \leq \theta_k \left\{((1 - \beta_{k-1}) + c_1 \beta_{k-1}) \|e_{k-1}\|_\infty\right\} + c_2 \cdot (\|\delta_k\|_\infty + \|\delta_{k-1}\|_\infty)|\tau_1| \|\delta_{k-1}\|_\infty$$
$$+ c_2 \cdot \sum_{i=2}^m \left(\|\delta_k\|_\infty + \|\delta_{k-i}\|_\infty + 2 \sum_{l=1}^{i-1} \|\delta_{k-i}\|_\infty\right) |\tau_i| \|\delta_{k-i}\|_\infty. \tag{7}$$

Note that the first term of RHS in Eq. (7) characterizes an increased convergence radius by an extra contraction factor $\theta_k$. Therefore, if the error terms $\|\delta_{k-i}\|_\infty$ are small enough, and the operator $\mathcal{T}$ is differentiable and has bounded first and second derivatives, the faster convergence result characterized by $\theta_k$ can be derived for the Q-value function iteration with Anderson acceleration.

Unfortunately, the commonly used max operator in $\mathcal{T}$ does not satisfy Assumption 1 as it is not a differentiable operator. Moreover, the "hard" max operator in $\mathcal{T}$ always commits to the maximum action-value function according to current estimation for updating the value estimator, lacking the ability to consider other potential action-values. A natural alternative is the Boltzmann Softmax operator, but this operator is prone to misbehave [3] as it is not a non-expansive operator. MellowMax operator [3], which can help strike a balance between exploration and exploitation, is considered in this paper. More importantly, the more meaningful convergence result of Anderson acceleration in policy iteration under Assumption 2 can be established due to the contraction properties of the Bellman operator under MellowMax operator. The results are given in Theorem 2.

**Assumption 2.** *Assume the Bellman operator $\mathcal{T}$ acting on state-action value function $Q$ is a $\gamma$-contraction operator, i.e., $\|\mathcal{T}Q - \mathcal{T}Q'\|_\infty \leq \gamma \|Q - Q'\|_\infty$ for each state-action function pair $Q$ and $Q'$.*

**Theorem 2.** *If both Assumption 1 and 2 hold, the coefficients $\alpha_i^{(k)}$ remain bounded and away from zero. The following bound holds for the residual $e_k$ with depth $m$*

$$\|e_k\|_\infty \leq \theta_k[(1-\beta_{k-1}) + c_1\beta_{k-1}]\|e_{k-1}\|_\infty + O(\sum_{j=1}^m \|e_{k-j}\|_\infty^2). \tag{8}$$

From Theorem 2, we find that there is a theoretical advantage to consider Anderson acceleration for policy iteration with depth $m$ due to the gain $\theta_k$ even with the higher-order accumulating terms. Fortunately, the Bellman operator with MellowMax operator is a $\gamma$-contraction operator, satisfying Assumption 2. Specifically, the resulting Bellman operator $\mathcal{T}_{mm}$ under the MellowMax operator is defined as

$$\mathcal{T}_{mm}Q(s,a) = R(s,a) + \gamma \sum_{s'\in S} p\left(s' \mid s,a\right) mm_\omega \left(Q\left(s',\cdot\right)\right), \tag{9}$$

where $mm_\omega$ is the MellowMax operator and $mm_\omega Q\left(s',\cdot\right) = \log(\frac{1}{|\mathcal{A}|}\sum_{a'}\exp[\omega Q(s',a')])/\omega$. Rigorously, we show that MellowMax can simultaneously satisfy Assumption 1 and 2 in Appendix B. As such, the faster convergence result presented in Theorem 2 can be derived. In other words, the faster convergence of policy iteration under MellowMax operator can be established by applying Anderson acceleration. Based on this theoretical principle, we apply MelloxMax operator in the Bellman operator to design our method in Section 3, where a detailed discussion is also provided.

Finally, the $Q$-value estimate $Q^{(k)}$ can be obtained by iteratively applying the MellowMax operator by starting from some initial value $Q^{(0)}$:

$$Q^{(k+1)} \leftarrow (1-\beta_k)\sum_{i=0}^m \alpha_i^{(k)}Q^{(k-m+i)} + \beta_k\sum_{i=0}^m \alpha_i^{(k)}\mathcal{T}_{mm}Q^{(k-m+i)}, \quad \forall(s,a)\in(\mathcal{S},\mathcal{A}). \tag{10}$$

## 3 Stabilization Analysis and Our Method

In this section, a stable regularization is firstly introduced and its stability analysis is provided as well. We then briefly analyze the role that MellowMax operator plays when conducting Anderson acceleration on deep reinforcement learning. These two factors eventually inspire our algorithm, which we call *Stable Anderson Acceleration (Stable AA)*.

### 3.1 Stable Regularization

Inspired by recent stable results of Anderson acceleration [9], we introduce the stable regularization term on the aforementioned unconstrained optimization problem Eq. (5) to obtain mixing coefficients $\tau^k$. Particularly, we add $\ell_2$ regularization of $\tau^k$ scaled by the Frobenius norm of $\Delta_k$ and $H_k$ to improve the stability. This yields the new optimization problem

$$\tau^k = \underset{\tau\in\mathbb{R}^m}{\operatorname{argmin}} \|e_k - H_k\tau\|^2 + \eta\left(\|\Delta_k\|_F^2 + \|H_k\|_F^2\right)\|\tau\|^2, \tag{11}$$

where $\eta$ is a positive tunning parameter representing the scale of regularization. The solution is $\tau^k = (H_k^T H_k + \eta(\|\Delta_k\|_F^2 + \|H_k\|_F^2)\mathrm{I})^{-1}H_k^T e_k$. We introduce this stable regularization under the unconstrained variables $\tau^k$, which facilitates the optimization. Intuitively, if the algorithm converges, we have $\lim_{k\to\infty}\|\Delta_k\|_F = \lim_{k\to\infty}\|H_k\|_F = 0$. Therefore, the coefficient on the regularization term vanishes as the algorithm converges, degenerating to Anderson acceleration method without the stable regularization. In this sense, the stability owing to our stable regularization plays a more important role in the early phase of training, which we demonstrate in Section 4. Based on the solved stable regularization $\tau^k$ and the relationship between $\tau^k$ and $\alpha^{(k)}$, we derive the updating of $Q^{(k+1)}$ in the policy iteration as follows

$$Q^{(k+1)} = Q^{(k)} - \tilde{G}_k e_k, \tag{12}$$

where $\tilde{G}_k = -\beta_k I + (\Delta_k + \beta_k H_k)\left(H_k^T H_k + \eta(\|\Delta_k\|_F^2 + \|H_k\|_F^2)\mathrm{I}\right)^{-1}H_k^T$.

Moreover, the following Theorem characterizes the stability ensured by regularization in Eq (11). Please refer to the proof in Appendix C.

**Theorem 3.** *The matrix $\tilde{G}_k$ satisfy $\|\tilde{G}_k\|_2 \le |2/\eta - \beta_k|$, $\|\tilde{G}_k^{-1}G_k\|_2 < 1$.*

Theorem 3 derives the upper bound of $\|\tilde{G}_k\|_2$, which is determined by $\eta$ and $\beta_k$. Intuitively, a larger strength of regularization $\eta$ and a proper magnitude of $\beta_k$ can yield more stability. In addition, $\|\tilde{G}_k^{-1}G_k\|_2$ is strictly less than 1, revealing a smaller violation in $Q^{(k)}$ iteration compared with non-regularized form.

To quantify the effect of regularization on the coefficient $\alpha^{(k)}$, we provide some analytical results regarding the obtained mixing coefficients $\alpha^{(k)}$ in Proposition 2. The proof is provided in Appendix C.

**Proposition 2.** *Let $\alpha_{non}^{(k)}$ and $\alpha_{reg}^{(k)}$ be the mixing coefficient vectors obtained by vanilla unconstrained and our stable regularized Anderson acceleration, respectively. Define the transformation matrix as $A$, satisfying $\alpha^{(k)} = A \cdot \tilde{\tau}^k$ with $\tilde{\tau}^{(k)} = (1, \tau^k)^T$ (detailed structure of $A$ is in the Appendix C). Let $cond_2(A)$ be the conditional number of $A$, i.e., $cond_2(A) = \|A\|_2 \|A^{-1}\|_2$. Then we have the following inequalities*

$$\|\alpha_{reg}^{(k)}\|_2^2 \le 4\left(1 + \frac{\|e_k\|^2}{\eta^2}\right), \quad \|\alpha_{reg}^{(k)} - \alpha_{non}^{(k)}\|_2^2 \le (cond_2(A))^2 \cdot \left\|\alpha_{non}^{(k)}\right\|_2^2 - \frac{2m+1}{m+1}. \tag{13}$$

From the first inequality, we observe that the $\ell_2$-norm of the derived coefficients $\alpha_{reg}^{(k)}$ is controlled by the regularization parameter $\eta$. An overly large $\eta$ tends to reduce the bound for the norm of $\alpha_{reg}^{(k)}$, implying a stable solution of the mixing coefficients $\alpha_{reg}^{(k)}$. Besides, we can conclude from the second inequality that there is an inevitable gap between $\alpha_{non}^{(k)}$ and $\alpha_{reg}^{(k)}$.

### 3.2 Stability Effects of MellowMax

The adopted MellowMax operator bridges the Anderson acceleration and reinforcement learning algorithms and it has two-sided stability effects. Firstly, based on the convergence analysis in Section 2, MellowMax operator satisfies the *differential and non-expansive* properties, which allows the faster convergence of Anderson acceleration in policy iteration. In contrast, the commonly used max and Boltzmann Softmax operator [4, 28] violate one of the theoretical assumptions respectively, and thus the (faster) convergence of Anderson acceleration under them may not be guaranteed. This is likely to yield instability while the training of algorithms.

Secondly, it is well-known that the "hard" max updating scheme in the popular off-policy methods, such as Q-learning [32], may lead to misbehavior due to the overestimation issue in the noisy environment [29, 13, 21]. By contrast, it has been demonstrated that MellowMax and Softmax operators are capable of reducing the overestimation biases, therefore reducing the gradient noises to stabilize the optimization of neural networks [3, 27]. The stable gradient estimation based on MellowMax operator leads to the enhancement of final performance for algorithms.

### 3.3 Algorithm: Stable AA

The introduced stable regularization approach combined with the MellowMax operator finally form our Stable AA method. In our algorithm, we focus on exploring the impact of Stable AA on Dueling-DQN [31]. In particular, under the procedure of off-policy learning in DQN, we firstly formulate the general damped Anderson acceleration form with the function approximator $Q_\theta$ as follows

$$Q_\theta(s_t, a_t) = \beta_t \sum_{i=1}^{m} \hat{\alpha}_i Q_{\theta^i}(s_t, a_t) + (1 - \beta_t)\mathbb{E}_{s_{t+1}, r_t}\left[r_t + \gamma \sum_{i=1}^{m} \hat{\alpha}_i \max_{a_{t+1}} Q_{\theta^i}(s_{t+1}, a_{t+1})\right],$$
$$\tag{14}$$

where $\theta_i$ are parameters of target network before the $i$-th update step. $\hat{\alpha}_i$ can be computed either by vanilla Anderson acceleration [11], or Regularized Anderson acceleration [25]. Then the obtained $Q_\theta(s_t, a_t)$ serves as the target in the updating of Q-networks. In our Stable AA method, we firstly solve the optimization problem in Eq. (11) to compute $\tau^k$. Next we obtain $\tilde{\alpha}^{(k)}$ by making use of the quantitative relationship between $\tau^k$ and $\alpha^{(k)}$. More importantly, we substitute max with MellowMax operator $mm_\omega$. The resulting target value function $\widetilde{Q}_\theta$ in our Stable AA algorithm is

---

**Algorithm 1** Stable AA Dueling-DQN Algorithm

---
1: Initialize a Q value network $Q_\theta$ and $m$ target networks with parameters $\theta_i$ ($i = 1, ..., m$). Set the total training steps $K$ and updating step $M$.
2: **while** $k \leq K$ **do**
3:     Observe the initial state $s_0$;
4:     **for** $t = 1$ to $T$ **do**
5:         Select $a_t = \arg\max_a Q_\theta(s_t, a)$ with probability $1 - \epsilon$ and a random action with probability $\epsilon$.
6:         Perform the action $a_t$, obtain $r_t$ and $s_{t+1}$. Store the transition $(s_t, a_t, r_t, s_{t+1})$ in the replay buffer.
7:         Sample the batch of transitions $(s, a, r, s')$ from the replay buffer.
8:         / * *Step 1: compute $\tilde{\alpha}^{(k)}$ * /*
9:         Compute $\Delta_k$ and $H_k$, and then solve the optimization problem with stable regularization in Eq. (11) to obtain $\tau^k$.
10:        Obtain the optimal coefficient vectors $\tilde{\alpha}^{(k)}$ via $\tilde{\alpha}^{(k)} = A \cdot \tilde{\tau}^k$, where the transformation matrix $A$ is defined in Proposition 2.
11:        / * *Step 2: compute the target $\widetilde{Q}_\theta$ by Anderson Mixing * /*
12:        Compute the value after the MellowMax operator for each target network $Q_{\theta_i}$, i.e., $mm_\omega(Q_{\theta_i}(s_{t+1}, \cdot))$
13:        Evaluate the target value function $\widetilde{Q}_\theta(s_t, a_t)$ via Eq. (15) under the Melloxmax operator.
14:        / * *Step 3: update the Q value networks * /*
15:        Update the Q value network $\theta$ by minimizing the loss in Eq.(16) with the target $y_t$ from Step 2.
16:        Update $m$ target networks every $M$ steps, i.e., $\theta_i \leftarrow \theta_{i+1}(i = 1, ..., m)$ and $\theta_m \leftarrow \theta$.
17:        Set $k = k + 1$.
18:     **end for**
19: **end while**

---

reformulated as

$$\widetilde{Q}_\theta(s_t, a_t) = \beta_t \sum_{i=1}^m \widetilde{\alpha}_i Q_{\theta^i}(s_t, a_t) + (1 - \beta_t)\mathbb{E}_{s_{t+1}, r_t}\left[r_t + \gamma \sum_{i=1}^m \widetilde{\alpha}_i \cdot mm_\omega(Q_{\theta^i}(s_{t+1}, \cdot))\right],$$

(15)

where $mm_\omega(Q(s, \cdot))$ is the MellowMax operator. Finally, the updating is performed by minimizing the least squared errors of Bellman equation between the current Q value estimate $Q_\theta(s_t, a_t)$ and the target value function $y_t$ obtained from Eq. (15),

$$L(\theta) = \mathbb{E}_{(s_t, a_t) \in \mathcal{D}}\left[(y_t - Q_\theta(s_t, a_t))^2\right],$$

(16)

where $\mathcal{D}$ is the distribution of previously sampled transitions.

In summary, the key of Stable AA method in policy iteration lies in two factors: the stable regularization in Eq. (11) in computing coefficient $\alpha^{(k)}$, and the MellowMax operator enabling the faster convergence in updating $Q^{(k)}$, both of which improve the convergence and sample efficiency. Moreover, we provide a detailed description of Stable AA on Dueling-DQN algorithm in Algorithm 1. Similar to the strategy in [25], the incorporation of Stable AA into policy gradient based algorithms, including actor critic [28] and twin delayed DDPG (TD3) [10] can be directly implemented in their critics part. It can be viewed as a straightforward extension, and we leave this exploration as future works.

## 4 Experiment

Our theoretical results about Anderson acceleration mainly apply to the case of a tabular value function representation, but our derived Stable AA algorithms can be naturally applied into the function approximation setting. The goal of our experiments is to show that our Stable AA method can still be useful in practice by improving the performance of Dueling-DQN algorithms. Our experimental results demonstrate that such an improvement is attributed to the joint benefits of the proposed stable regularization and the MelloMax operator.

**Experimental Settings**  We perform our Stable AA Dueling-DQN algorithm on a variety of Atari games, and mainly focus on reporting four representatiave games, i.e., SpaceInvaders, Enduro, Breakout, and CrazyClimber. Results of other games are similar, which we provide in Appendix D. We compare our approach with Dueling-DQN [31] and Regularized Anderson Acceleration (RAA)[25]. In addition, we also provide an ablative analysis about our Stable AA algorithm to illuminate the separate and joint impacts of stable regularization and MellowMax operator. In the following experiments, we report results statistics by running three independent random seeds. We set $\beta_t$ in Eq. (15) as 0.1 for convenience.

**Implementation of MellowMax Operator**  The MellowMax operator $mm_\omega$ satisfies the desirable differential and non-expansive properties, enabling the faster convergence in policy iteration with Anderson acceleration. Nevertheless, we need to perform an additional root-finding algorithm [3] to compute the optimal $\omega$ in each state in order to maintain these properties and help the MellowMax operator to identify a probability distribution. Unfortunately, this root-finding algorithm is computationally expensive to be applied. Following the strategy in [27], we tune the inverse parameter $\omega$ from $\{1, 5, 10\}$ and then report the best score.

## 4.1 Performance of Our Stable AA

We select DuelingDQN and DuelingDQN-RAA as baselines for the evaluation on the four Atari games. These two algorithms and our approach DuelingDQN-Stable AA are trained under the same random seeds and evaluated every 10,000 environment steps, where each evaluation reports the average returns with standard deviations. Our implementation is adapted from RAA [25]. After the grid search, we set $\omega$ in MelloxMax of Stable AA as 5.0 and $\eta$ in stable regularization as 0.1 across 4 Atari games.

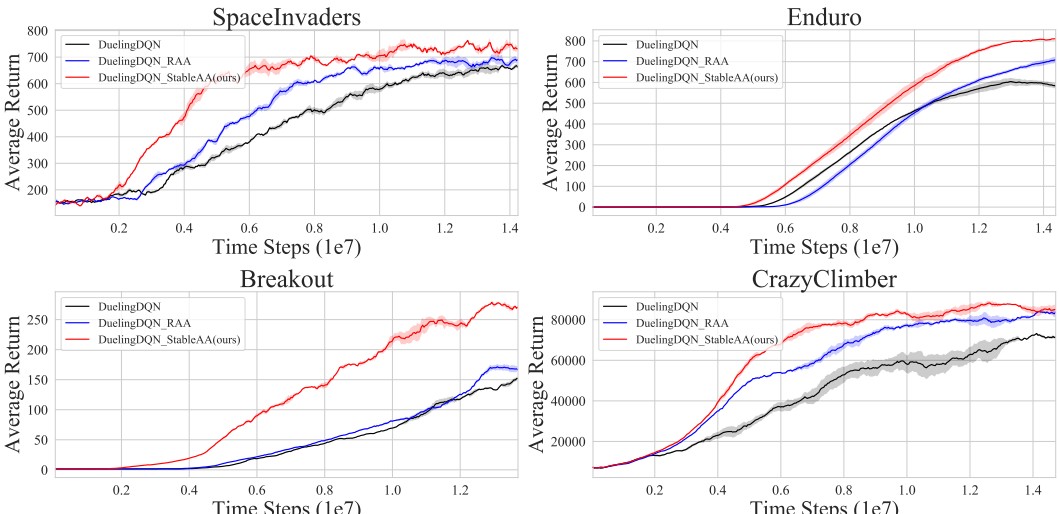

Figure 1: Learning curves of DuelingDQN, DuelingDQN-RAA and our approach DuelingDQN-Stable AA on SpaceInvaders, Enduro, Breakout, and CrazyClimber games over 3 seeds. Shaded region corresponds to the standard deviation.

From Figure 1, we note that our DuelingDQN-StableAA (red line) significantly outperforms Regularized AA (blue line) and baseline (black line) across all four games. Overall, Dueling-RAA enables to accelerate DuelingDQN to improve the sample efficiency and enhance the final performance, but our approach can lead to further benefits. Remarkably, our DuelingDQN-StableAA (red line) is superior to RAA to a large margin, especially on Breakout where our approach achieves around 250 average return while RAA only achieves 150 return. In summary, we conclude that the joint impact of both stable regularization and theoretically-principled MellowMax further accelerate the convergence and improve the sample efficiency of the popular off-policy DuelingDQN algorithm.

## 4.2 Ablation Analysis

We further examine the separate impact of the proposed stable regularization (shown in Eq. (11)) and the theoretically-principled MelloxMax operator via the rigorous ablation study. Starting from DuelingDQN, we firstly add stable regularization with different scales $\eta$ while comparing with our resulting Stable AA method. Meanwhile, we separately replace Max operator in DuelingDQN with MellowMax operators with various inverse parameters $\omega$ to explore their impacts.

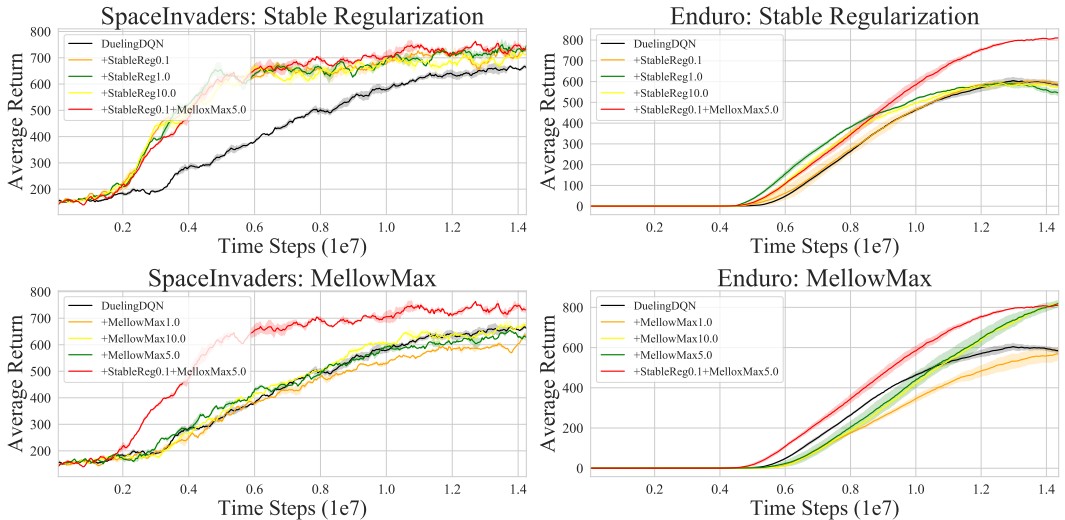

Figure 2: Learning curves of DuelingDQN, +MellowMax, +Stable Regularization (+StableReg), and +Stable Regularization+MellowMax on SpaceInvaders, Enduro games over 3 seeds.

**Impact of Stable Regularization**  From diagrams in the first row of Figure 2, we can observe that the benefit margin of stable regularization on Anderson acceleration differs from game to game. Concretely, naively applying stable regularization on DuelingDQN regardless of the MellowMax to guarantee the faster convergence of Anderson acceleration can still significantly accelerate the convergence in Spaceinvaders. In contrast, the stable regularization is able to boost the sample efficiency mildly on Enduro. For example, when $\eta = 1.0$ (green line), "+StableReg1.0" is more sample efficient (higher than yellow and orange lines) in the early phase of training. However, the benefit of stable regularization vanishes as the training proceeds, achieving comparable performance with DuelingDQN. Interestingly, if we further add the additional MellowMax operator (red line), the resulting Stable AA approach can accomplish the improvement of performance to a large margin.

**Impact of MellowMax Operator**  As exhibited in diagrams in the last row of Figure 2, the benefit of MellomMax operator still depends on the game. Particularly, the improvement of MellowMax operator on SpaceInvaders is negligible, where the lines representing "+MelloxMax" overlap subtly with DuelingDQN. Nevertheless, our Stable AA additionally incorporates the stable regularization, achieving remarkable improvement of sample efficiency. In addition, due to the fact that it is hard to compute the optimal inverse temperature $\omega$ in MellowMax, we tune the parameter $\omega$ and report the corresponding results in Figure 2. It manifests from the diagram on Enduro game that MellowMax under $\omega = 5.0, 10.0$ (green and yellow lines) can substantially enhance the final performance. More importantly, under the joint benefits of both the stable regularization and the theoretically-principled MellowMax operator, our Stable AA DuelingDQN algorithm can simultaneously accelerate the convergence and improve the final performance.

## 5  Discussion and Conclusion

Apart from MellowMax, other variants of Softmax operator can also be combined with Anderson acceleration, although their theoretical principles have not been studied. For instance, the competitive

performance of the Boltzmann Softmax operator suggests that it is still preferable in certain domains, despite its non-contraction property. We leave the exploration towards more desirable operators as future works. Additionally, the study of our approach on a wider variety of Atari games, and implement on more state-of-the art algorithms are expected in the future.

In this paper, we firstly provide deeper insights into the mechanism of Anderson acceleration on the reinforcement learning setting by connecting damped Anderson acceleration with quasi-Newton method and providing the faster convergence results. These theoretical principles about the faster convergence of Anderson acceleration inspire the leverage of MellowMax operator. Combing with a stable regulation, the resulting Stable AA strategy is applied in DuelingDQN, which has been further demonstrated to significantly accelerate the convergence and enhance the final performance.

## Acknowledgement

We would like to thank the anonymous reviewers for great feedback on the paper. Yingnan Zhao and Ke Sun were supported by the State Scholarship Fund from China Scholarship Council (No:202006120405 and No:202006010082). Dr. Jiang and and Dr. Kong were supported by the Natural Sciences and Engineering Research Council of Canada (NSERC). Dr. Kong was also supported by the University of Alberta/Huawei Joint Innovation Collaboration, Huawei Technologies Canada Co., Ltd., and Canada Research Chair in Statistical Learning.

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
