# A  Proof: Connection with quasi-Newton

*Proof.*

$$Q^{(k+1)} = (1 - \beta_k) \sum_{l=0}^{m} \alpha_l^{(k)} Q^{(k-m+l)} + \beta_k \sum_{l=0}^{m} \alpha_l^{(k)} TQ^{(k-m+l)}$$

$$= (1 - \beta_k) \left[ Q^{(k)} - \sum_{i=0}^{m-1} \tau_i \left( Q^{(k-i)} - Q^{(k-i-1)} \right) \right]$$

$$\quad + \beta_k \left[ T_{mm} Q^{(k)} - Q^{(k)} + Q^{(k)} - \sum_{i=0}^{m-1} \tau_i \left( TQ^{(k-i)} - T_{mm} Q^{(k-i-1)} \right) \right]$$

$$= Q^{(k)} - (1 - \beta_k) \Delta_k \cdot \tau - \beta_k \cdot (\Delta_k + H_k) \tau + \beta_k e_k$$

$$= Q^{(k)} + \beta_k e_k - (\Delta_k + \beta_k H_k) \tau$$

$$= Q^{(k)} - ((\Delta_k + \beta_k H_k) \left( H_k^T H_k \right)^{-1} H_k^T - \beta_k I) e_k$$

$$:= Q^{(k)} - G_k e_k$$

This formula indicates the term $G_k = (\Delta_k + \beta_k H_k) \left( H_k^T H_k \right)^{-1} H_k^T - \beta_k I$ can be seen as the inverse Jacobian of $e_k = TQ^{(k)} - Q^{(k)}$. $\qquad\square$

# B  Proof in Convergence results

**Proof about Assumption 1** This proof is to show that MellowMax operator satisfies Assumption 1.

*Proof.* We first show $T_{mm}$ is twice continuously differentiable. For any vector $x = (x_1, \ldots, x_n)^T$, we have

$$\frac{\partial mm_\omega(x)}{\partial x_i} = \frac{\exp(\omega x_i)}{\sum_l \exp(\omega x_l)}$$

and

$$\frac{\partial mm_\omega(x)}{\partial x_i^2} = \frac{\omega \exp(\omega x_i) \left[ \sum_l \exp(\omega x_l) \right] - (\exp(\omega x_i))^2 \omega}{\left( \sum_l \exp(\omega x_l) \right)^2}$$

$$\frac{\partial mm_\omega(x)}{\partial x_i \partial x_j} = \frac{\partial mm_\omega(x)}{\partial x_\partial x_i} = \frac{-\omega \exp(\omega(x_i + x_j))}{(\sum_l \exp(\omega x_l))^2},$$

which implies that $T_{mm}$ is twice continuously differentiable due to smoothness $\exp(\cdot)$ and for any bounded domain in $\mathbb{R}^n$, the first and second order derivative exist.

We next show the first and second derivative of $T_{mm}$ are bounded which follows from $\|T_{mm}(Q + \Delta) - T_{mm}Q\|_\infty \leq c_1 \|\Delta\|_\infty + c_2 \|\Delta^2\|_\infty + o(\|\Delta^2\|_\infty)$ for any $\Delta \to 0$.

$$\|T_{mm}(Q + \Delta) - T_{mm}Q\|_\infty = \left\| \frac{\gamma}{\omega} \cdot P \cdot \log \{I \exp[\omega(Q + \Delta)]\} - \frac{\gamma}{\omega} \cdot P \cdot \log \{I \exp(\omega Q)\} \right\|_\infty$$

$$= \left\| \frac{\gamma}{\omega} \cdot P \cdot \log \{I \exp(\omega \Delta)\} \right\|_\infty$$

$$= \left\| \frac{\gamma}{\omega} \cdot P \cdot \left[ (I \exp(\omega \Delta) - I) - \frac{1}{2} (I \exp(\omega \Delta) - I)^2 + o(\Delta^2) \right] \right\|_\infty$$

$$\leq \left\| \frac{\gamma}{\omega} \cdot P \cdot [I \exp(\omega \Delta) - I] \right\|_\infty + o(\|\Delta^2\|_\infty)$$

$$= \left\| \frac{\gamma}{\omega} \cdot P \cdot \left[ I\omega\Delta + \frac{1}{2}I\Delta^2\omega^2 \right] \right\|_\infty + o(\|\Delta^2\|_\infty)$$

$$\leq c_1\|\Delta\|_\infty + c_2\|\Delta^2\|_\infty + o(\|\Delta^2\|_\infty), \tag{A1}$$

$$\square$$

where $P = [p(s_{i'}|s_i, a_j]_{1 \leq i, i' \leq |\mathcal{S}|, 1 \leq j \leq m}$.

**Proof about Theorem 1** This proof is to show that we have the results in Theorem 1 under Assumption 1.

*Proof.* Let

$$Q_k^\alpha(s,a) = \sum_{l=0}^m \alpha_l^{(k)} Q^{(k-m+l)}(s,a)$$

$$\widetilde{Q_k^\alpha}(s,a) = \sum_{l=0}^m \alpha_l^{(k)} T_{mm} Q^{(k-m+l)}(s,a)$$

Then

$$Q^{(k+1)}(s,a) = (1-\beta_k)Q_k^\alpha(s,a) + \beta_k \widetilde{Q^\alpha}(s,a).$$

Define $T'(\cdot;\cdot)$, $T''(\cdot;\cdot,\cdot)$ as linear form with respect to the arguments to the right of semicolon. Let $\delta_k = Q^{(k)} - Q^{(k-1)}$, $z_k(t) = Q^{(k-1)} + t\delta_k$, $z_{k,t}(u) = z_{k-1}(t) + u(z_k(t) - z_{k-1}(t))$. Then

$$
\begin{aligned}
T_{mm}(Q^{(k)}) - T_{mm}(Q^{(k-1)}) &= \int_0^1 T'_{mm}(z_k(t);\delta_k))dt \\
&= \int_0^1 \left\{ T'_{mm}(z_{k+1}(t);\delta_k) + \int_0^1 T''_{mm}(z_{k+1,t}(s);z_k(t)-z_{k+1}(t),\delta_k)ds \right\} dt \\
&= \int_0^1 \int_0^1 \left\{ T'_{mm}(z_{k+1}(t);\delta_k) + T''_{mm}(z_{k+1,t}(s);z_k(t)-z_{k+1}(t),\delta_k) \right\} dsdt.
\end{aligned}
$$

We note that

$$
\begin{aligned}
e_k = T_{mm}(Q^{(k)}) - Q^{(k)} &= T_{mm}(Q^{(k)}) - [(1-\beta_{k-1})Q_{k-1}^\alpha + \beta_{k-1}\widetilde{Q_{k-1}^\alpha}] \\
&= (1-\beta_{k-1})[T_{mm}(Q^{(k)}) - Q_{k-1}^\alpha] + \beta_{k-1}[T_{mm}(Q^{(k)}) - \widetilde{Q_{k-1}^\alpha}]
\end{aligned}
\tag{A2}
$$

For each term on the right hand of formula (A2), we have

$$
\begin{aligned}
T_{mm}Q^{(k)} - Q_{k-1}^\alpha &= \sum_{i=0}^m \alpha_i^{(k-1)} T_{mm}Q^{(k)} - \sum_{i=0}^m \alpha_i^{(k-1)} Q^{(k-m+i-1)} \\
&= \sum_{i=0}^m \alpha_i^{(k-1)}(T_{mm}Q^{(k)} - Q^{(k-m+i-1)}) \\
&= \sum_{i=0}^m \alpha_i^{(k-1)}(T_{mm}Q^{(k-m+i-1)} - Q^{(k-m+i-1)}) + \sum_{i=0}^m \alpha_i^{(k-1)}(T_{mm}Q^{(k)} - T_{mm}Q^{(k-m+i-1)}) \\
&= e_{k-1}^\alpha + \sum_{i=0}^m \left( \sum_{l=0}^{m-i} \alpha_l^{(k-1)} \right)(T_{mm}Q^{(k-i)} - T_{mm}Q^{(k-i-1)}) \\
&= e_{k-1}^\alpha + \sum_{i=0}^m \tau_i \widetilde{\delta_{k-i}},
\end{aligned}
$$

where $e_k^\alpha = \sum_{i=0}^m \alpha_i^{(k)}(T_{mm}Q^{(k-m+i)} - Q^{(k-m+i)})$, $\tau_i = \sum_{l=0}^{m-i} \alpha_l^{(k-1)}$, $\widetilde{\delta_{k-i}} = T_{mm}Q^{(k-i)} - T_{mm}Q^{(k-i-1)}$. Moreover,

$$
\begin{aligned}
T_{mm}Q^{(k)} - \widetilde{Q_{k-1}^\alpha} &= T_{mm}Q^{(k)} - \sum_{i=0}^m \alpha_i^{(k-1)} T_{mm}Q^{(k-i-1)} \\
&= \sum_{i=0}^m \alpha_i^{(k-1)}(T_{mm}Q^{(k)} - T_{mm}Q^{(k-i-1)}) \\
&= \sum_{i=0}^m \tau_i \widetilde{\delta_{k-i}}.
\end{aligned}
$$

Therefore, formula (A2) can be rewritten as

$$
e_k = (1 - \beta_{k-1})(e_{k-1}^\alpha + \sum_{i=0}^m \tau_i \widetilde{\delta_{k-i}}) + \beta_{k-1} \sum_{i=0}^m \tau_i \widetilde{\delta_{k-i}}
$$

$$
= (1 - \beta_{k-1})e_{k-1}^\alpha + \sum_{i=0}^m \tau_i \widetilde{\delta_{k-i}}
$$

$$
= (1 - \beta_{k-1})e_{k-1}^\alpha + \sum_{i=0}^m \tau_i \int_0^1 T'_{mm}(z_{k-i}(t); \delta_{k-i}) dt
$$

$$
= (1 - \beta_{k-1})e_{k-1}^\alpha + \sum_{i=1}^m \tau_i \left\{ \int_0^1 T'_{mm}(z_k(t); \delta_{k-i}) dt \right.
$$

$$
\left. + \sum_{l=k-i}^{k-1} \int_0^1 T'_{mm}(z_l(t); \delta_{k-i}) - T'_{mm}(z_{l+1}(t); \delta_{k-i}) dt \right\} + \int_0^1 T'_{mm}(z_k(t); \delta_k) dt
$$

$$
= (1 - \beta_{k-1})e_{k-1}^\alpha + \int_0^1 T'_{mm}(z_k(t); \sum_{i=0}^m \tau_i \delta_{k-i}) dt
$$

$$
+ \sum_{i=1}^m \tau_i \sum_{l=k-i}^{k-1} \int_0^1 \int_0^1 T''_{mm}(z_{l+1,t}(s); z_l(t) - z_{l+1}(t), \delta_{k-i}) ds dt
$$

$$
= (1 - \beta_{k-1})e_{k-1}^\alpha + \int_0^1 T'_{mm}(z_k(t); \sum_{i=0}^m \tau_i \delta_{k-i}) dt
$$

$$
+ \sum_{i=1}^m \int_0^1 \int_0^1 \sum_{l=k-i}^{k-1} T''_{mm}(z_{l+1,t}(s); z_l(t) - z_{l+1}(t), \tau_i \delta_{k-i}) ds dt.
$$

For the term $\sum_{i=0}^m \tau_i \delta_{k-i}$, it can be rewritten as

$$
\sum_{i=0}^m \tau_i \delta_{k-i} = \delta_k + \sum_{i=1}^m \tau_i \delta_{k-i}
$$

$$
= Q^{(k)} - Q^{(k-1)} + \tau_1 Q^{(k-1)} - \sum_{i=0}^{m-1} \alpha_i Q^{(k-m+i-1)}
$$

$$
= Q^{(k)} - \alpha_m^{(k-1)} Q^{(k-1)} \sum_{i=1}^{m-1} \alpha_i^{(k-1)} Q^{(k-m+i-1)}
$$

$$
= Q^{(k)} - Q_{k-1}^\alpha
$$

$$
= \beta_{k-1}(\widetilde{Q_{k-1}^\alpha} - Q_{k-1}^\alpha) = \beta_{k-1} e_{k-1}^\alpha,
$$

where the second and third equality hold using the formula $\tau_i - \tau_{i+1} = \alpha_{m-i}^{(k-1)}$, $\tau_1 = 1 - \alpha_m^{(k-1)}$. Then, we obtain

$$
e_k = \int_0^1 (1 - \beta_{k-1})e_{k-1}^\alpha + \beta_{k-1} T'_{mm}(z_k(t); e_{k-1}^\alpha) dt
$$

$$
+ \sum_{i=1}^m \int_0^1 \int_0^1 \sum_{l=k-i}^{k-1} T''_{mm}(z_{l+1,t}(s); z_l(t) - z_{l+1}(t), \tau_i \delta_{k-i}) ds dt. \quad \text{(A3)}
$$

Formula (A1) and (A3) together imply that

$$
\|e_k\|_\infty \leq (1 - \beta_{k-1})\|e_{k-1}^\alpha\|_\infty + \beta_{k-1} \cdot c_1 \cdot \|e_{k-1}^\alpha\|_\infty + \sum_{i=1}^m \sum_{l=k-i}^{k-1} c_2 \cdot (\|\delta_l\|_\infty + \|\delta_{l+1}\|_\infty) |\tau_i| \|\delta_{k-i}\|_\infty
$$

$$
= \theta_k \left\{ ((1 - \beta_{k-1}) + c_1 \beta_{k-1}) \|e_{k-1}\|_\infty \right\} + c_2 \cdot \sum_{i=2}^m \left( \|\delta_k\|_\infty + \|\delta_{k-i}\|_\infty + 2 \sum_{l=1}^{i-1} \|\delta_{k-i}\|_\infty \right) |\tau_i| \|\delta_{k-i}\|_\infty
$$

$$
+ c_2 \cdot (\|\delta_k\|_\infty + \|\delta_{k-1}\|_\infty) |\tau_1| \|\delta_{k-1}\|_\infty. \quad \text{(A4)}
$$

$\square$

**Proof about Assumption 2** This proof is to show that MellowMax operator satisfies Assumption 2 (non-expansive operator). Similar result is also given in [3, 14].

*Proof.* Let $|\mathcal{S}| = n_1$, $\mathcal{A} = n_2$. Note that

$$T_{mm}Q = R + \gamma \cdot \mathrm{P} \cdot mm_\omega(Q)$$

where $mm_\omega(Q) = \frac{1}{\omega} \log\{\frac{1}{n_2} \cdot \mathrm{I} \cdot \exp(\omega Q)\}$, $\mathrm{I} = \mathrm{I}_{n_1 \times n_1} \otimes 1_{n_2 \times 1}^T$.

$$\begin{aligned}
\|T_{mm}Q - T_{mm}Q'\|_\infty &\leq \gamma \|\mathrm{P}\|_\infty \|mm_\omega(Q) - mm_\omega(Q')\|_\infty \\
&\leq \gamma \|mm_\omega(Q) - mm_\omega(Q')\|_\infty \\
&\leq \gamma \|Q - Q'\|_\infty
\end{aligned} \tag{A5}$$

$\square$

**Proof about Theorem 2** We analyze a bound for $\delta_j$ in terms of $e_j$ in the following part. Based on formula (A5), we have

$$\begin{aligned}
(1-\gamma)\|\delta_k\|_\infty &= \|\delta_k\|_\infty - \gamma\|\delta_k\|_\infty \\
&\leq \|\delta_k\|_\infty - \|T_{mm}Q^{(k)} - T_{mm}Q^{(k-1)}\|_\infty \\
&\leq \left\| Q^{(k)} - Q^{(k-1)} - T_{mm}Q^{(k)} + T_{mm}Q^{(k-1)} \right\|_\infty \\
&= \|e_k - e_{k-1}\|_\infty.
\end{aligned} \tag{A6}$$

Let $E_k = (e_{k-m}, \ldots, e_k)$. The optimization problem

$$\alpha^k = \mathrm{argmin}_{\alpha \in \mathbb{R}^{m+1}} \|E_k \alpha\|_2^2 \quad s.t. \sum_{i=0}^m \alpha_i = 1$$

is equivalent to the unconstrained form

$$\min_{\eta \in \mathbb{R}^m} \quad \|e_{k-m} + \sum_{i=1}^m \eta_i(e_{k-m+i} - e_{k-h+i-1})\|^2, \quad \eta_i = \sum_{l=i}^m \alpha_l^{(k)} \tag{A7}$$

$$\min_{\tilde{\tau} \in \mathbb{R}^m} \left\| e_k - \sum_{i=0}^{m-1} \tilde{\tau}_i (e_{k-i} - e_{k-i-1}) \right\|^2, \quad \widetilde{\tau}_i = \sum_{l=0}^{m-i-1} \alpha_l^{(k)} \tag{A8}$$

Seeking the critical point for $\eta_m$ in (A7) yields that

$$\langle e_{k-m}, e_k - e_{k-1} \rangle + \sum_{i=1}^m \eta_i \langle e_{k-m+i} - e_{k-m+i-1}, e_k - e_{k-1} \rangle = 0.$$

This implies that

$$\begin{aligned}
\eta_m \|e_k - e_{k-1}\|^2 &= -\langle e_{k-m}, e_k - e_{k-1} \rangle - \sum_{i=1}^{m-1} \eta_i \langle e_k - e_{k-1}, \quad e_{k-m+i} - e_{k-m+i-1} \rangle \\
&= -\eta_{m-1}\langle e_k - e_{k-1}, e_{k-1} \rangle - \left\langle e_k - e_{k-1}, \sum_{i=0}^{m-2} \alpha_i e_{k-m+i} \right\rangle.
\end{aligned}$$

Applying Cauchy-Schwarz inequality and triangle inequalities yields

$$\left| \alpha_m^{(k)} \right| \|e_k - e_{k-1}\| \leq |\eta_{m-1}| \|e_{k-1}\| + \sum_{i=0}^{m-2} \alpha_i^{(k)} \|e_{k-m+i}\|.$$

Based on the inequality $\|\cdot\|_\infty \leq \|\cdot\|_2$ over $\mathbb{R}^n$ and formula (A6), it follows

$$\left| \alpha_m^{(k)} \right| \|\delta_k\|_\infty \leq \frac{1}{1-\gamma} \left\{ |\eta_{m-1}| \|e_{k-1}\| + \sum_{i=0}^{m-2} \alpha_i^{(k)} \|e_{k-m+i}\| \right\}. \tag{A9}$$

Seeking the critical point with respect to $\tilde{\tau}_p$, $(p = 1, \ldots, m-1)$ in (A8) yields

$$\left\langle e_k - \sum_{i=0}^{m-1} \tilde{\tau}_i (e_{k-i} - e_{k-i-1}), \quad e_{k-p} - e_{k-p-1} \right\rangle = 0$$

which implies

$$\tilde{\tau}_p \|e_{k-p} - e_{k-p-1}\|^2 = \langle e_{k-p} - e_{k-p-1}, \tilde{\tau}_{p-1}e_{k-p}\rangle - \langle e_{k-p} - e_{k-p-1}, \tilde{\tau}_{p+1}e_{k-p-1}\rangle$$

$$+ \left\langle e_{k-p} - e_{k-p-1}, \sum_{j=0}^{m-p-2} \alpha_j e_{k-m+j}\right\rangle + \left\langle e_{k-p} - e_{k-p-1}, \sum_{j=m-p+1}^{m} \alpha_j e_{k-m+j}\right\rangle.$$

Then

$$|\tilde{\tau}_p| \|\delta_{k-p}\|_\infty \leq \frac{1}{1-\gamma} \left\{ |\tilde{\tau}_{p-1}| \|e_{k-p}\| + |\tilde{\tau}_{p+1}| \|e_{k-p-1}\| + \sum_{j=0}^{m-p-2} |\alpha_j| \|e_{k-m+j}\| + \sum_{j=m-p+1}^{m} |\alpha_j| \|e_{k-m+j}\| \right\}$$

$$\text{(A10)}$$

Combing (A4), (A9) and (A10), we establish

$$\|e_k\|_\infty \leq \theta_k \left\{ ((1-\beta_{k-1}) + c_1\beta_{k-1}) \|e_{k-1}\|_\infty \right\} + Constant \cdot \left\{ \sum_{i=2}^{m} \|\delta_{k-i}\|_\infty^2 + \|\delta_{k-1}\|_\infty^2 \right\}$$

$$= \theta_k \left\{ ((1-\beta_{k-1}) + c_1\beta_{k-1}) \|e_{k-1}\|_\infty \right\} + O\left( \sum_{i=1}^{m} \|e_{k-i}\|_\infty^2 \right).$$

# C    Proof: Stable regularization

We firstly prove the stability of the derived regularization in Theorem 3.

*Proof.* It is easy to prove that $\|\tilde{G}_k^{-1} G_k^{-1}\|_2 \leq 1$ as long as we directly remove the regularization term to induce the inequality. Then, we have

$$\|\tilde{G}_k\|_2 \leq \left| -\beta_k + \frac{\|\Delta_k + \beta_k H_k\|_2 \|H_k\|_2}{\eta \left(\|\Delta_k\|_F^2 + \|H_k\|_F^2\right)} \right|$$

$$\leq \left| -\beta_k + \frac{\|\Delta_k\|_2 \|H_k\|_2 + \|H_k\|_2^2}{\eta \left(\|\Delta_k\|_F^2 + \|H_k\|_F^2\right)} \right|$$

$$\leq \left| -\beta_k + \frac{\|\Delta_k\|_F \|H_k\|_F + \|H_k\|_F^2}{\eta \left(\|\Delta_k\|_F^2 + \|H_k\|_F^2\right)} \right|$$

$$\leq \left| \frac{2}{\eta} - \beta_k \right|.$$

$\square$

This indicates that the $\ell_2$ norm of updating matrix $\tilde{G}_k$ is upper bounded, which can guarantee the stability. Then we provide the proof of Proposition 2.

*Proof.* Firstly, we denote the structure matrix A as follows:

$$A = \begin{pmatrix} 0 & 0 & 0 & 0 & \cdots & 0 & 0 & 1 \\ 0 & 0 & 0 & 0 & \cdots & 0 & 1 & -1 \\ \vdots & \vdots & \vdots & \vdots & \vdots & \vdots & \vdots & \vdots \\ 0 & 0 & 1 & -1 & \cdots & 0 & 0 & 0 \\ 0 & 1 & -1 & 0 & \cdots & 0 & 0 & 0 \\ 1 & -1 & 0 & 0 & \cdots & 0 & 0 & 0 \end{pmatrix}_{(m+1)\times(m+1)}$$

Note that $\alpha_{\text{reg}}^{(k)} = A \cdot \tilde{\tau}_{\text{reg}}$, where

$$\tilde{\tau}_{\text{reg}} = \begin{pmatrix} 1 \\ \tau_{\text{reg}} \end{pmatrix} \in \mathbb{R}^{m+1}.$$

We first bound $\alpha_{\text{reg}}^{(k)}$,

$$\|\alpha_{\text{reg}}^{(k)}\|_2^2 \leq \|A\|_2^2 \cdot \|\widetilde{\tau}_{\text{reg}}\|_2^2 \leq 4 \cdot \left(1 + \|\tau_{\text{reg}}\|_2^2\right)$$

$$\leq 4\left[1 + \left\|\left(H_k^\top H_k + \eta\left(\|\Delta_k\|_F^2 + \|H_k\|_F^2\right)I\right)^{-1}\right\|^2 \cdot \|H_k^T e_k\|^2\right]$$

$$\leq 4\left(1 + \frac{\|H_k^T e_k\|^2}{\eta^2\left(\|\Delta_k\|_F^2 + \|H_k\|_F^2\right)}\right)$$

$$\leq 4\left(1 + \frac{\|e_k\|^2}{\eta^2}\right).$$

We next analyze $\alpha_{\text{reg}}^{(k)} - \alpha_{\text{non}}^{(k)}$. Since

$$H_k^T e_k - H_k^T H_k \tau_{\text{non}} = 0,$$
$$H_k^T e_k - \left[H_k^T H_k + \eta\left(\|\Delta_k\|_F^2 + \|H_k\|_F^2\right)I\right]\tau_{\text{reg}} = 0$$

Then $\tau_{\text{reg}} - \tau_{\text{non}} = \left[H_k^T H_k + \eta\left(\|\Delta_k\|_F^2 + \|H_k\|_F^2\right)I\right]^{-1}\left[\eta\left(\|\Delta_k\|_F^2 + \|H_k\|_F^2\right)I\right]\tau_{\text{non}}$ which implies

$$\|\tau_{\text{reg}} - \tau_{\text{non}}\|_2 \leq \frac{\left(\eta\left(\|\Delta_k\|_F^2 + \|H_k\|_F^2\right)\right)\|\mathbf{I}\|_2}{\eta\left(\|\Delta_k\|_F^2 + \|H_k\|_F^2\right)}\|\tau_{\text{non}}\|_2 = \|\tau_{\text{non}}\|_2. \tag{A11}$$

Let $\tilde{\tau}_{\text{reg}} = (1, \tau_{\text{reg}}^T)^T$, $\tilde{\tau}_{\text{non}} = (1, \tau_{\text{non}}^T)^T$. Then $\tilde{\tau}_{\text{non}} = A^{-1}\alpha_{\text{non}}^{(k)}$, and $\|\tau_{\text{non}}\|_2^2 = \|A^{-1}\alpha_{\text{non}}^{(k)}\|_2^2 - 1$. Based on (A11), we can establish

$$\|\alpha_{\text{reg}}^{(k)} - \alpha_{\text{non}}^{(k)}\|_2^2 \leq \|A\|_2^2\|\tilde{\tau}_{\text{reg}} - \tilde{\tau}_{\text{non}}\|_2^2 = \|A\|_2^2\|\tau_{\text{reg}} - \tau_{\text{non}}\|_2^2$$

$$\leq \|A\|_2^2\left(\left\|A^{-1}\alpha_{\text{non}}^{(k)}\right\|_2^2 - 1\right)$$

$$\leq \|A\|_2^2 \cdot \left\|A^{-1}\right\|_2^2 \cdot \left\|\alpha_{\text{non}}^{(k)}\right\|_2^2 - \|A\|_2^2$$

$$\leq (cond_2(A))^2 \cdot \left\|\alpha_{\text{non}}^{(k)}\right\|_2^2 - \frac{2m+1}{m+1}.$$

$\square$

Figure 3: Learning curves of DuelingDQN, DuelingDQN-RAA, DuelingDQN-Stable AA (ours) on Aline, BattleZone, Berzerk and Bowling games over 3 seeds.

# D   Results on Other games

We provide results of our algorithms on other 12 Atari games. Our results in Figure 3,4,5 and 6 show that our Stable AA DuelingDQN consistently outperforms both DuelingDQN and DuelingDQN-RAA significantly.

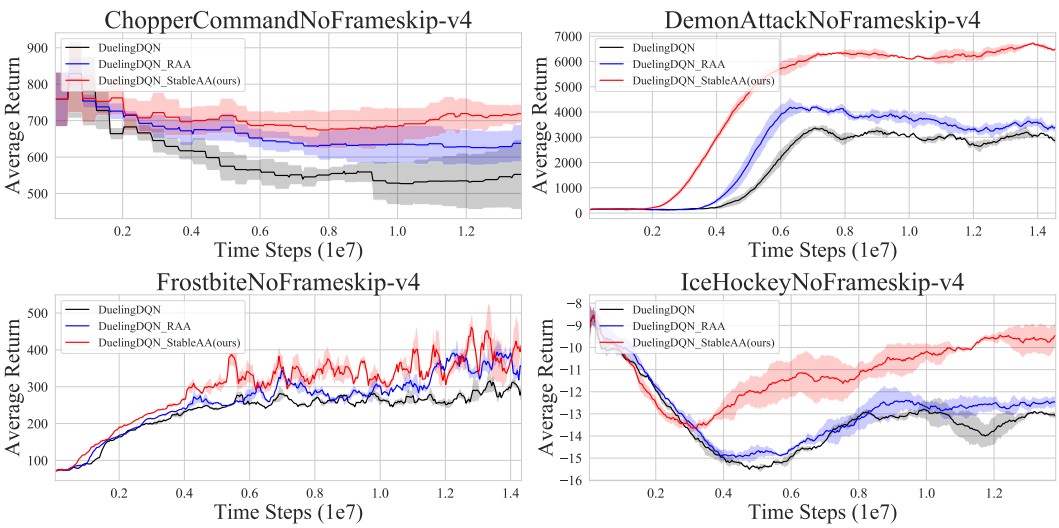

Figure 4: Learning curves of DuelingDQN, DuelingDQN-RAA, DuelingDQN-Stable AA (ours) on ChopperCommand, DemonAttack, Frostbite and IceHockey games over 3 seeds.

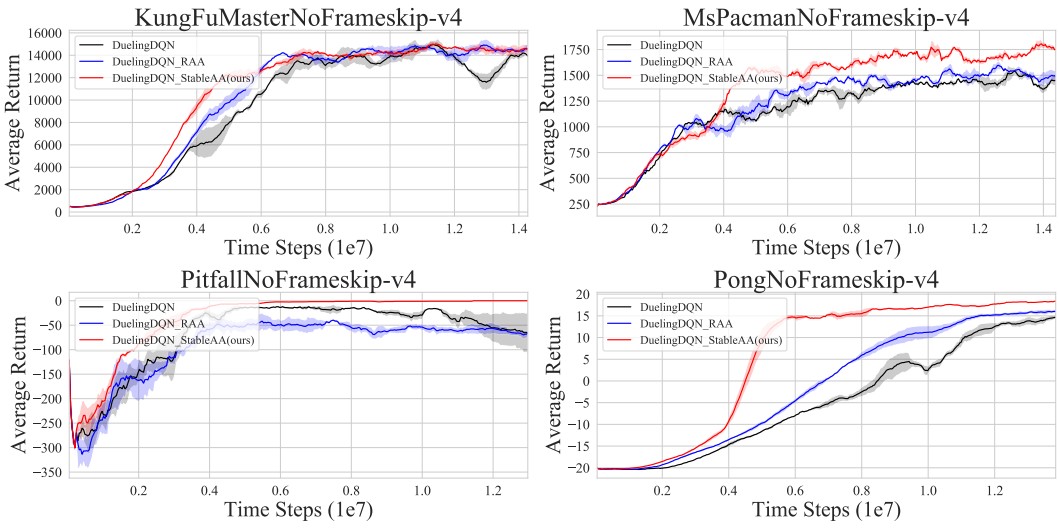

Figure 5: Learning curves of DuelingDQN, DuelingDQN-RAA, DuelingDQN-Stable AA (ours) on KungFu, MsPacman, Pitfall and Pong games over 3 seeds.

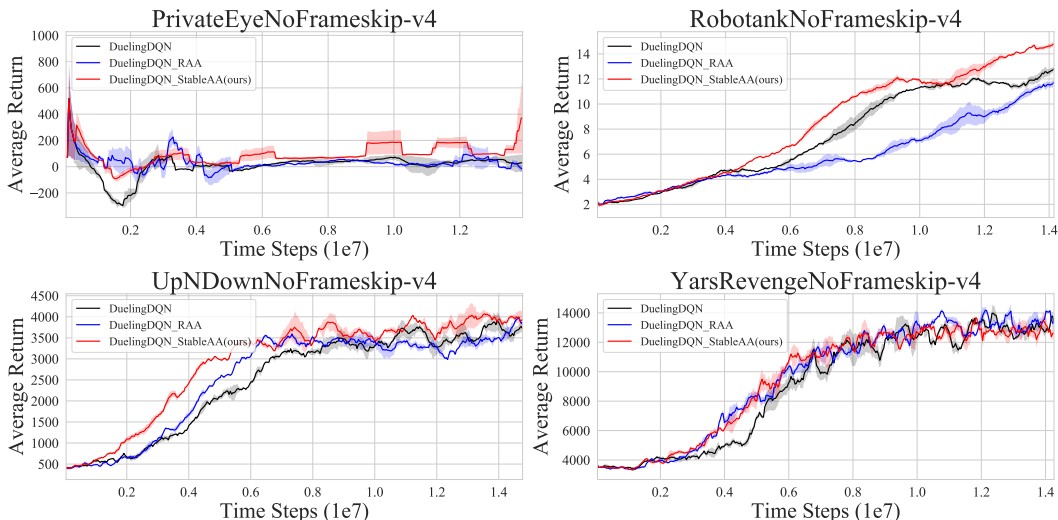

Figure 6: Learning curves of DuelingDQN, DuelingDQN-RAA, DuelingDQN-Stable AA (ours) on PrivateEye, Robotank, UpNDown and YarsRevenge games over 3 seeds.