# OpenReview forum: "Damped Anderson Mixing for Deep Reinforcement Learning: Acceleration, Convergence, and Stabilization"
_NeurIPS.cc/2021/Conference — NeurIPS 2021 Poster_

### Official Review · Reviewer_p6FQ · 2021-07-16

**Rating:** 7
**Confidence:** 3

**Summary:**

This paper looks to characterise the use of Anderson acceleration in Deep Q learning.  Theoretical convergence results are given in tabular settings. A novel algorithm is developed to satisfy the assumptions made in the theory. The algorithm outperforms reasonable baselines and ablation studies.

**Limitations And Societal Impact:**

The authors provide a reasonable characterisation of the limitations of their work. Societal impacts are not included, though the work doesn't have obvious social implications.

**Main Review:**

# Strengths
The paper provides strong theoretical results in characterising the convergence benefits of Anderson acceleration in tabular settings, and backs them up empirically, with good ablation studies. Assumptions are explained clearly, and the analysis is novel in the context of RL.

# Weaknesses
As far as I can tell, the algorithmic contribution here is limited, substituting the traditionally employed max operator for the already well-understood mellowmax operator.

The result obtained is relatively niche, as use of Anderson mixing is not standard practice in RL.

The connection between Anderson mixing and quasi-Newton's method has been established, though the application to RL provided here is insightful.

The tabular setting is restrictive.

# Correctness

The paper is correct throughout, and the empirical methodology appears to be sound. Three seeds is generally very few for RL settings, though the algorithm seems to perform consistently across those three seeds.

# Clarity

The paper is easy to read and straightforward, though the presentation of empirical results is somewhat confusing. The difference between the RAA baseline and the ablation studies is not clear to me.

# Relation to Prior Work
The paper is well placed in the context of acceleration methods as applied to RL, and makes use of contemproary results from the existing literature on Anderson mixing.

# Reproducability
The work seems reasonably reproducible. Hyperparameters are reported.




**Time Spent Reviewing:**

4h

---

> ### Author Response · Authors · 2021-08-10
> **Response to Reviewer p6FQ**
>
> Thank you for your constructive feedback and valuable comment. Please kindly find the detailed responses below.
>
> $\textbf{Q1. Novelty and contribution}$
>
> As mentioned by other reviewers, our main contribution lies in providing the mathematical insights of Anderson mixing in RL by establishing its connection with quasi-Newton as well as obtaining the convergence rate. In addition, we propose a novel stabilization strategy by introducing a regularization term in AA and the theoretically-principled MellowMax operator, followed by its stability analysis. Extensive experiments demonstrate the effectiveness of our proposed algorithm in a wide range of environments.
>
> $\textbf{Q2. Clarity}$
>
> The main differences between our algorithm and RAA are the introduction of our stable regularization and the theoretically principled MellowMax operator. Therefore, the goal of our ablation studies is to demonstrate that the improvement of our proposed algorithm is attributed to the joint benefits of the stable regularization and the MelloMax operator. Please refer to Section 4.2 for more explanation.

---

> > ### Comment · Reviewer_p6FQ · 2021-09-01
> > **Thanks**
> >
> > Thanks to the authors for their response. I remain convinced that the work is suitable for publication. Thanks for the clarifying comments.

---

### Official Review · Reviewer_8vvP · 2021-07-16

**Rating:** 7
**Confidence:** 4

**Summary:**

Anderson mixing is a heuristic applied to RL to improve sample-efficiency and convergence. This paper delves into the theoretical side of the Anderson mixing that improves the convergence of deep RL algorithms, showing that the Anderson mixing adds an extra contraction. This paper also shows the connection between Anderson mixing and Mellowmax operator, showing more stable performance and fast convergence.

**Ethics Review Area:**

["I don’t know"]

**Main Review:**

- I think the introduction part is very well-written; background explanation about GSVI, JVI, average-DQN, and the context of using Anderson mixing was helpful.
- Section 2.1: Anderson acceleration: linear combination of m previous estimates of Q-values, regulated by the damping parameter \beta_k
- I think connection between damped Anderson acceleration and quasi-Newton method is nicely presented.
- Mellowmax operator fulfills the two assumptions made in Assumption 1, 2. I think the non-expansiveness of the operator is already given in the previous work. Can you give a citation on the Proof of Assumption 2 (Asadi et al. 2017) ?
- Stable regularization: tau^k contains Frobenius norm delta_k and H_k, which is tuned by the parameter \eta.
- Stable AA algorithm = stable regularization approach combined with Mellowmax operator. (1) computes delta_k and H_k for the optimization problem to obtain \tau_k, (2) target value obtained using Mellowmax, (3) updates target networks.
- Experiments comparing Dueling-DQN and Dueling-DQN-RAA is provided in multiple Atari game domains. Ablation studies comparing the effect of regularization and mellowmax parameters are also given.

I think this paper is a well-written paper with sound theoretical analysis. The connection between Anderson mixing and stable regularization, and their combination with a Mellowmax operator is the main key point of the paper. I think the originality of the paper is somewhat overlapping with the previous work on Anderson mixing and RL (Shi et al. 2019), but the main strength of this paper comes with the analysis. Experimental results in Atari games were solid.

Questions

Q1. There are m target networks in Stable AA Dueling-DQN algorithm. However, I couldn't find detailed information on how many target networks the authors used in practice. How does the choice of m affect the performance of the algorithm? How did you select m? How does it affect the computation and memory?

Q2. It seems that the range of Mellowmax inverse temperature parameter \omega {1,5,10} is somewhat limited. And there doesn't seem to be a big performance change depending on the choice of \omega in SpaceInvaders and Enduro. Can we argue that the temperature sensitivity is low in this case? could you add more experiments using a very large inv temperature parameter? e.g. \omega=100, 1000?

Citation suggestions

- I hope you add the following citation to the non-expansiveness proof of the appendix (although it is already cited in this paper)

[1] K. Asadi and M. L. Littman. An alternative softmax operator for reinforcement learning. ICML 2017

- Overestimation alleviation (section 3.2.) of Mellowmax (proof):

[2] S.Kim, K. Asadi, M.L. Littman, G. D. Konidaris. DeepMellow: Removing the Need for a Target Network in Deep Q-learning. IJCAI 2019.

**Time Spent Reviewing:**

8

---

> ### Author Response · Authors · 2021-08-10
> **Response to Reviewer 8vvP**
>
> Thank you for your constructive feedback and valuable comment. Below are our responses.
>
> $\textbf{Q1. Impart of $m$ target networks}$
>
> Following [1], the number $m$ of target networks is set to 5 across all Atari games. The sensitivity analysis shows that a larger $m$ leads to faster convergence and better final performance, but the improvement becomes small when $m$ exceeds a threshold, which was also indicated in [1]. In addition, a larger $m$ slightly increases the computation and thus we suggest taking into account available computing resources and sample efficiency when applying our proposed method.
>
> $\textbf{Q2. $\omega$ in MellowMax}$
>
> The reason that we select best $\omega$ among \{1, 5, 10\} as [2] is we observed an overly large $\omega$ would significantly decrease the performance of algorithms. For a clear comparison, we conduct the experiment on Breakout as an example over 3 seeds, and the final performance is presented here.
>
> •DuelingDQN: 152.26±9.32
>
> •DuelingDQN+RAA: 166.61±10.8
>
> •DuelingDQN+StableReg+MellowMax1000:  199.89±6.46
>
> •DuelingDQN+StableReg+MellowMax100:  232.18±27.16
>
> •DuelingDQN+StableReg+MellowMax5:  271.05±5.29
>
> $\textbf{Q3. Citation suggestions}$
>
> Thank you for this suggestion and we will add them in the revised version.
>
> $\textbf{Reference}$
>
> [1] Shi, Wenjie, et al. Regularized Anderson acceleration for off-policy deep reinforcement learning. (NeurIPS 2019)
>
> [2] Song, Zhao, Ron Parr, and Lawrence Carin. Revisiting the softmax bellman operator: New benefits and new perspective. (ICML 2019)

---

### Official Review · Reviewer_5Yxv · 2021-07-17

**Rating:** 7
**Confidence:** 3

**Summary:**

The paper provides a mathematical justification for the benefits of Anderson Accelerating in RL by connecting Anderson mixing and quasi-newton method. The paper also propose a stable Anderson Accelerating (stable AA )algorithm with a stable regularization term and a MellowMax operator and evaluate the performance with Dueling DQN on a variety of Atari games environments. Experimental results show that the proposed algorithm improves convergence, stability and final performance compared to vanilla Dueling DQN and Dueling DQN with previous RAA method.

**Limitations And Societal Impact:**

cons:
1. In addition to Dueling DQN, I'm interested in how stableAA will perform when combined with TD3. (I think in the RAA paper, the RAA algorithm has been used with TD3 to evaluate on continuous Mujoco environments.) If we can see similar performance gains on both TD3 and Dueling-DQN comparing to. previous work, the conclusion would be more promising and meaningful.

**Main Review:**

 Originality: The idea of introducing Anderson acceleration to the context of RL is not new, but the as the authors mentions, mathematical insights for the benefits needs more study. The authors bridge the gap by introducing a connection between Anderson mixing and quasi-Newton. While the originality is somewhat limited in using AA in DRL, developing a new algorithm combining AA and MellowMax operator which needs better performance is not trivial.

Quality: The paper is technically sound, and the experimental analysis is fair and supports the main thesis of the paper. The improvements  improves both the learning speed and final performance over Dueling DQN and Dueling DQN with RAA on some Atari games environments.

Clarity: The paper is clearly written. The background on Anderson acceleration and the proofs are clearly stated, as well as the motivation and extension to the deep RL domain.

Significance: This work proposes a modification to existing off-policy algorithm. The results support the promise of moderate performance gains. The established connection between Anderson mixing and quasi-Newton methods bridge the gap of mathematical analysis in the field.

**Time Spent Reviewing:**

40

---

> ### Author Response · Authors · 2021-08-10
> **Response to Reviewer 5Yxv**
>
> Thank you for your constructive feedback and valuable comment. Please kindly find the detailed responses below.
>
> $\textbf{Q1. About extension to TD3}$
>
> We appreciate this suggestion. The extension of our stable AA to policy gradient-based algorithms, such as TD3, is valuable.  We conduct a preliminary experiment of the extension to TD3 on the Hopper environment. Below are the comparison results across 7 random seeds.
>
> •TD3:  2281.81±1738.93
>
> •TD3+RAA: 1707.91±1353.94
>
> •TD3+StableAA (ours):  3180.28±695.66
>
> It turns out that our proposed regularization strategy performs best compared with baseline and RAA on the Hopper environment.  As the MelloxMax operator may not be directly applicable to TD3, we only apply the stable AA regularization in our experiment,  which also demonstrates its superiority.  We leave more rigorous comparison experiments about the extension of our algorithm as future works.

---

### Decision · Program_Chairs · 2021-09-27

**Decision:**

Accept (Poster)

**Comment:**

The paper is a nice mix of theoretical and experimental results. It analyzes Anderson mixing in RL, finding a link with quasi-newton methods and providing a convergence rate. The assumptions of the analysis hold for the MellowMax operator (introduced in published work). Fairly thorough experiments on Atari with ablations provide evidence for the usefulness of stable Anderson acceleration and the mellowmax operator.